# Attraction and Avoidance between Predators and Prey at Wildlife Crossings on Roads

**Cristina Mata** [1,2,*] , **Jesús Herranz** [1,2] **and Juan E. Malo** [1,2]

1    Terrestrial Ecology Group (TEG-UAM), Departamento de Ecología, Facultad de Ciencias, Universidad Autónoma de Madrid. C Darwin 2, 28049 Madrid, Spain; jesus.herranz@uam.es (J.H.); je.malo@uam.es (J.E.M.)
2    Centro de Investigación en Biodiversidad y Cambio Global (CIBC-UAM), Universidad Autónoma de Madrid. C Darwin 2, 28049 Madrid, Spain
*    Correspondence: cristina.mata@uam.es

**Abstract:** Wildlife passages are currently built at roads and railway lines to re-establish connectivity. However, little is known about whether predator-prey interactions may reduce the effectiveness of the crossing structures. We evaluated the co-occurrence patterns of predator-prey species-pairs at 113 crossing structures, noting their coincidence at the same structure and/or on the same day. We built occupancy models using presence-absence matrices for three prey and five predator types obtained during 2076 passage-days of monitoring. The results indicate that predators and prey do not use passages independently. Attraction or segregation effects occurred in 20% of predator-prey species-pairs and were detected in 67% of cases with respect to same-day use. Our results show that both predator and prey species used the same structures to cross fenced roads. However, the spatial and daily patterns of crossing suggest that there were predators that attended crossings to search for prey and that prey species avoided using crossings in the presence of predators. Our results support two recommendations to avoid crossing structures losing effectiveness or becoming prey traps: (i) increase the number of wider structures to reduce the risks of predator-prey encounters and (ii) include inside them structural heterogeneity and refuges, to reduce the likelihood for predator-prey interactions.

**Keywords:** co-occurrence; motorway; occupancy modeling; prey trap; road ecology; species interactions

## 1. Introduction

Road and railway constructions bring about not only habitat destruction but also fragment the landscape into smaller and more isolated patches. At a regional extent, this entails a transformation of ecosystems, involving changes in their composition, structure, and functioning and in the populations of the species that inhabit them [1–4]. Division of habitats into isolated fragments has been associated with various changes in biotic interactions, including those between predators and their prey [5,6]. Thus, re-establishing connectivity within a landscape fragmented by built infrastructure may prove decisive in restoring the balance of predator-prey systems, since predator and prey populations are often interdependent, and landscape structure plays a major role in their dispersion and dynamics [5].

Wildlife passages are common measures taken to mitigate the impact of roads and railways in an attempt to reconnect landscape-scale ecological processes. They include bridges, tunnels, or adapted drainage culverts that serve to reduce roadkill and to reestablish both occasional dispersive faunal movements and regular daily or seasonal ones across the landscape [7]. Passages are generally designed

for and used by a great diversity of species, thus potentially bringing about interactions between predators and prey that might ultimately influence their effectiveness as mitigation measures [8,9].

Such predator-prey interactions might influence the effectiveness of wildlife passages if they result in interspecific exclusion or convert the crossings into predation traps. On the one hand, if crossing structures used by predators are avoided by prey [10], they may only resolve the fragmentation problems of the former. It is known that both predator and prey species routinely use wildlife crossings [11,12], but a partial temporal or spatial exclusion of prey from them due to predator presences may occur, and it may have passed undetected in analyses of vertebrate uses of passages not focused to analyzing the issue. On the other hand, it is possible that predators visit crossings in search of prey, thus increasing the risk of death for prey species. The passages may thus act as predation traps for them [13] or even as population sinks, as it has been described for a population of southern brown bandicoots by fox predation at a wildlife underpass [14]. Some authors have suggested the use of wildlife crossings as hunting sites [11,15], but few scientific studies have addressed the question [16]. Dickson and colleagues [17] found no evidence of cougars being attracted to crossing structures to hunt deer, and Ford and Clevenger [18] have shown that evidence of predator-prey interactions among large mammals at wildlife crossings do not match the prey-trap hypothesis in the Trans-Canada Highway. However, recently, Caldwell and Klip [19] have found that predator-prey interactions influenced underpass uses by several species, but little is known about the generality of these findings and, particularly, if passages may influence interactions between smaller-sized predators and prey. Among them, a close association between both groups may happen, and some species may show less aversion to the road environment [18].

In this study, we formally evaluated the existence of changes in the use of wildlife passages by predator and prey species as a response to the presence of each other. In case of no behavioral response to each other, it is expected that the use of passages by predators and prey species will fit the null hypothesis of spatial and temporal independency. On the contrary, if predators do visit passages in search of prey, a higher than expected occurrence of both together may happen. Alternatively, if prey avoid passages used by predators, the observed frequency of both together will be lower than expected. In case of the interdependent use of crossings by predators and prey, the functionality of such structures for wildlife could be impaired, and the extent of the problem should be addressed. It is relevant to note that co-occurrence or avoidance may happen in space and/or time and that the detection of coincident patterns at both scales can be especially interesting, since spatial coincidences could result just from similar habitat selections.

## 2. Materials and Methods

### 2.1. Crossing Structures and the Study Area

A total of 113 crossing structures were monitored along a 110-km-stretch of the A-52 motorway from Benavente (Zamora Province) to Orense (Orense Province), NW Spain during 2001–2003 (Figure S1). The monitored structures comprised 58 drainage culverts, 29 underpasses, and 26 overpasses (12 of them wildlife-specific; see more details in Table S1). The monitoring period varied among structures from 10–50 days, and 36% were monitored for 20 days or longer.

Monitoring involved the daily recording in the early morning of tracks within a one-meter-wide strip of marble dust extending from side to side across each crossing structure [9]. It was not always possible to identify tracks to a species level, making it necessary to classify them according to species groups comprising species of similar ecological roles and sizes. The basic datum included in the analyses was the daily presence/absence of each species (or group) in each pass irrespective of the actual number of tracks per day.



### 2.2. Prey and Predator Types Monitored Using Crossing Structures

The faunal groups studied comprised three prey types and five predator types, covering the full size-range of the most widespread mammals of European ecosystems, with the exception of ungulates and the brown bear, *Ursus arctos*:

(a) Prey: small mammals (mice, voles, and shrews); rat-sized rodents, hereafter "rats" (*Arvicola sapidus, A. terrestris, Rattus rattus,* and *R. norvegicus*); and lagomorphs (*Oryctolagus cuniculus* and *Lepus granatensis*).

(b) Predators: small mustelids (weasel *Mustela nivalis* and stoat *M. erminea*), cats (*Felis catus* and *F. silvestris*), Eurasian badger *Meles meles*, red fox *Vulpes vulpes*, and large canids (*Canis familiaris* and *C. lupus*).

In general, all predators in this study prey on rodents and lagomorphs, although the extent to which they do it varies [20–29].

### 2.3. Analysis of Co-occurrence Patterns

Co-occurrence patterns were analyzed for all possible predator-prey species-pairs by employing presence-absence matrices based on the 2076 passage days monitored at the 113 structures. The dataset comprised 2329 records: 52.2% prey species and 47.8% predators (Table 1).

**Table 1.** Sample size of the data used to build co-occurrence models by species and passage type. n = the number of structures used by each species. d = the number of days on which each species was recorded. *n* = the total number of each structure type monitored.

| Prey and Predator Species | Culverts (*n* = 58) | | Underpasses (*n* = 29) | | Overpasses (*n* = 26) | | Total (*n* = 113) | |
|---|---|---|---|---|---|---|---|---|
| | n | d | n | d | n | d | n | d |
| *Prey* | | | | | | | | |
| Small mammals | 51 | 508 | 23 | 125 | 22 | 184 | 96 | 817 |
| Rats | 15 | 43 | 6 | 16 | 2 | 2 | 23 | 61 |
| Lagomorphs | 13 | 28 | 22 | 177 | 18 | 133 | 53 | 338 |
| *Predators* | | | | | | | | |
| Small mustelids (*Mustela* spp.) | 16 | 50 | 6 | 6 | 0 | 0 | 22 | 56 |
| Eurasian badger | 14 | 69 | 14 | 85 | 2 | 3 | 30 | 157 |
| Cats (*Felis* spp.) | 22 | 50 | 15 | 48 | 13 | 28 | 50 | 126 |
| Red fox | 34 | 102 | 27 | 201 | 21 | 81 | 82 | 384 |
| Large canids (*Canis* spp.) | 33 | 77 | 27 | 177 | 24 | 136 | 84 | 390 |

Interdependency between species was calculated with statistical occupancy models [30], which are ideal for matrices with unequal monitoring periods. These models included two fundamental parameters that allow testing for the probability that the detection of one species depends on the presence of the other, due to spatial (at the same passage) or temporal (on the same day) coincidences. Since more than 90% of wildlife crossings take place between sunset and sunrise (Mata unpublished data), the coincidence in one day is in fact a measure of events taking place during the previous night, even though the species order of crossing and the time lapse between them remain unknown.

Following the terminology established by Mackenzie and colleagues [31], the two parameters comprise the following species interaction factors (SIF):

(1) $\varphi$: a parameter related to potential predator-prey interactions at individual structures (Equation (1)). Spatial co-occurrence hereafter.

$$\varphi = \Psi^{AB}/(\Psi^A * \Psi^B) \tag{1}$$

where $\Psi^{AB}$ is the probability that both species are present at one location (passage), $\Psi^A$ is the probability that species A is present at the location, $\Psi^B$ is the probability of species B being present

at the location, and φ is the ratio of how likely both species co-occur at that location compared to what would be expected if they co-occurred at random.

(2)  δ: a parameter informing on potential species interactions during the same-day use of one crossing structure (Equation (2)). Temporal co-occurrence hereafter.

$$\delta = r_j^{AB}/(r_j^A * r_j^B) \tag{2}$$

where $r_j^{AB}$ is the probability of coincidence of both species during survey (day) *j* in any precise location, $r_j^A$ is the marginal probability of detecting species *A* across locations during survey *j*, and $r_j^B$ is the marginal probability of detecting species *B* across locations during survey *j*.

Parameters φ and δ act as multipliers for the probability of the appearance of each species relative to expectations under randomness, and thus, they reflect the potential interaction between both of them. Therefore, their value is 1 if there is no interaction, whereas values of φ or δ greater than 1 show that species tend to co-occur in space or time more frequently than expected. Conversely, values less than 1 point to species avoidance in the same structure or on the same day.

Passage type was included as a covariate in the models to control for its effect on the frequency of use in order to achieve a better model fit [30]. The analysis of the interactions in the pair small mammals-small mustelids only considered culverts, given that the limited use made of other structure types precluded model fitting. For the same reason, the analysis of the interactions between rats and their potential predators excluded overpasses. To simplify the presentation of the results, the parameters corresponding to the passage type covariates are not shown, given that they inform on the relative frequency of passage use by individual species but not on their interactions. More comprehensive information about the differential use of passage types by vertebrates can be found in Mata and colleagues [9,32].

We used the two-species analysis module of PRESENCE 2.4. [33] to fit the models. We developed 4 models for each predator-prey species-pair:

(a)  Limited model: without interactions (omits parameters φ and δ). It represents the spatial and temporal independence of species occurrence.

(b)  Partial interaction Model I: the spatial independence of species occurrence (omits interaction parameter φ).

(c)  Partial interaction Model II: temporal independence (omits interaction parameter δ).

(d)  Total interaction model (including both interaction parameters φ and δ): it represents species interdependence both in the use of a particular passage and in the same-day use of the same crossing.

Model comparison was based on Akaike's information criterion. Afterwards, we used standard likelihood ratio tests [31,34] to compute the probabilities associated to parameters φ and/or δ included in the most informative models (see more details in Table S2). As each prey species dataset was tested with the five potential predators, we applied Bonferroni sequential probability correction to these tests [35]. The nature of species interactions were established from parameters φ and δ estimated in the full interaction model.

To avoid misinterpretation, mathematical interaction parameters will be referred to as interdependencies between animal species from here on.

## 3. Results

It was possible to adjust the models corresponding to the 15 pairs of predator-prey interactions correctly, including both the spatial use parameter (φ, Equation (1)) and the temporal use parameter (δ, Equation (2)) (see Table S2). There were four instances in which 100% coincidence of observations of the two members of a species-pair considered in one of the structure types did not allow the calculation of the standard error of the φ parameter. Overall, the high number of instances of significant interaction

parameters stands out (Tables 2–4), and most of them point to changes larger than 10% in co-occurrence likelihood. The use of particular structures deviated from chance (spatial interdependence) for approximately 20% of the predator-prey species-pairs. Dependence between species in the daily use of crossings occurred in approximately 67% of the species cases. Detailed results for the interactions between each prey type and its potential predators follows:

**Table 2.** Parameters of interspecific interactions between small mammals and predators representing the deviation from randomness of the probability of species co-occurrence in a crossing structure ($\varphi$) or on a given day ($\delta$). Standard errors are given in parentheses, except where not computed due to a lack of variability at some passage types. * $p < 0.05$ and ** $p < 0.01$.

| Prey-Predator Pairs | | Small Mustelid | Cat | Eurasian Badger | Red Fox | Large Canids |
|---|---|---|---|---|---|---|
| Small mammal | $\varphi$ (SE) | 1.04(-) | 1.05(0.03) | 1.11(0.04) ** | 1.02(0.02) | 1.02(0.03) |
| | $\delta$ (SE) | 1.14(0.10) ** | 1.01(0.15) ** | 1.15(0.18) ** | 0.77(0.07) ** | 0.76(0.07) ** |

**Table 3.** Parameters of interspecific interactions between rat-sized rodents and predators representing the deviation from randomness of the probability of species co-occurrence in a crossing structure ($\varphi$) or on a given day ($\delta$). Standard errors are given in parentheses, except where not computed due to a lack of variability at some passage types. * $p < 0.05$ and ** $p < 0.01$.

| Prey-Predator Pairs | | Small Mustelid | Cat | Eurasian Badger | Red Fox | Large Canid |
|---|---|---|---|---|---|---|
| Rat-sized rodent | $\varphi$(SE) | 0.74(0.37) ** | 0.71(0.27) | 1.02(13.4) | 1.09(-) | 1.13(0.16) |
| | $\delta$(SE) | 0.86(0.36) * | 0.96(0.48) | 0.34(0.51) | 0.65(0.24) ** | 0.51(0.2) |

**Table 4.** Parameters of interspecific interactions between lagomorphs and predators representing the deviation from randomness of the probability of species co-occurrence in a crossing structure ($\varphi$) or on a given day ($\delta$). Standard errors are given in parentheses, except where not computed due to a lack of variability at some passage types. * $p < 0.05$ and ** $p < 0.01$.

| Prey-Predator Pairs | | Small Mustelid | Cat | Eurasian Badger | Red Fox | Large Canid |
|---|---|---|---|---|---|---|
| Lagomorph | $\varphi$(SE) | 0.99(0.08) | 0.93(0.13) | 1.59(0.21) ** | 1.07(-) | 1.19(-) |
| | $\delta$(SE) | 0.51(0.26) | 0.74(0.18) ** | 2.25(0.52) * | 1.12(0.12) ** | 1.34(0.15) |

### 3.1. Small Mammals

Small mammal prey models showed significant interdependencies in all cases, revealing co-occurrence and avoidance (Table 2). The probability of spatial co-occurrence recording badgers at passages also used by small mammals was significantly greater than chance (10.5%), and the same trend was detected in the same-day use patterns at structures, the probability of detecting the badger-small mammals pair increasing by 15.3% above chance. Temporal co-occurrence was also greater than expected for the small mustelid-small mammal pair (14.3%) and very slightly so for the cat-small mammal pair (1.4%). In contrast, there was a 23 percent lower than expected chance for co-occurrence between the small mammals-red fox pair and small mammals-large canids pair. In both these pairs, no significant interactions were detected in the spatial co-occurrence.

### 3.2. Rats

Rat-sized prey significantly avoided the use of structures used by small mustelids and foxes (Table 3). They were 25.5% less likely to use structures used by small mustelids. Furthermore, in relation with the temporal co-occurrence, the chance of rats and small mustelids presences occurring at a crossing on the same day was 13.6% less than expected. Similarly, the probability of detecting the fox and rat pair on the same day was 34.6% less than expected by chance.

### 3.3. Lagomorphs

In three out of five cases, models for lagomorphs showed significant interdependencies, revealing both co-occurrence and avoidance (Table 4). Badgers co-occurred significantly with lagomorphs, both spatially and temporally (59% and 125% greater than expected by chance, respectively). Similarly, the red fox-lagomorph pair showed 11.9% higher coincidence in the same-day use of structures than expected. Conversely, the temporal co-occurrence of lagomorphs with cats was 26% less than expected.

## 4. Discussion

Our results show that, although both prey species and predators use the same structures [11,12], their spatial and temporal patterns when using them to cross roads diverge from chance, suggesting that some predators attend wildlife crossings in search of prey and that some prey species avoid using the crossings in the presence of predators [10,16]. The ample dataset (over 100 sites and 2000 site-day observations) and the inclusion of a broad range of prey and predator species underpin the consistency of the observed patterns and the potential relevance of the results obtained, even though they are not definitive evidence of the process due to the correlative nature of the study. A further element in our results points in the same direction: the systematic consistency between spatial and temporal co-occurrence. The results of our analyses are thus more parsimoniously explained globally in terms of the existence of prey-detection and active prey-seeking behavior by predators and of mechanisms of predator avoidance by prey, rather than in a case-by-case search of plausible explanations for interdependencies. Changes in the probability of same-day use brought about by the presence of another species might lead, in the most extreme cases, to the total avoidance of a passage in a case of segregation or total coincidence in a case of co-occurrence. However, the detected patterns are not that extreme, and they lie in a gradient from positive values, reflecting active prey-seeking by a predator, to negative ones, in which the prey acts effectively to reduce its chances of meeting a predator [36].

An overview of the predator-prey species-pairs points to co-occurrence happening in those pairs that most probably interact, i.e., in those cases where predation intensity is likely to be greatest due to size matching. Although the predators included in this study are somewhat generalists, small mammals appear most frequently in the diets of small predators and lagomorphs in those of larger predators. Accordingly, instances of the co-occurrence of small mammals involve the smaller predators (mustelids and cats), and those of avoidance involve the larger ones, the red fox and canids. Conversely, the larger prey species considered (lagomorphs) show co-occurrence with larger predators and segregation with the smaller ones. Risk perception and the development of antipredator behaviors by prey species have frequently been studied, prey responses varying according to context, such as location and/or the predator involved [37]. Many animals distinguish between predators, discerning the threat levels posed and adjusting their responses accordingly (the threat-sensitive predator hypothesis [38,39]). Such studies have involved small mammals [40], rats [41], and lagomorphs [42] and have shown that these species alter their behaviors and habitat uses in the presence of predators, seeking areas where they feel safest [43,44]. A common finding of studies that analyze antipredator responses is the observation that prey avoid sites scent-marked by a predator [45].

Predator-prey interdependences in wildlife passage uses show some cases of avoidance behaviors by prey species. Thus, rat-sized rodents responded most intensely when confronted with small mustelids, one of their potential predators for whom such rodents comprise their main prey [21]. Rats and water voles clearly avoid using structures on the same days as mustelids and may even avoid using the same passages, reflecting a degree of both temporal and spatial segregation between them. These same rodents show a clear tendency to avoid using the same passages and the same day as foxes, possibly indicating avoidance [46], as shown under experimental conditions for *Rattus norvegicus* [47]. Our results also reveal a degree of avoidance of cats by lagomorphs, which is in accordance with studies demonstrating feline scents or feces repelling lagomorphs [48,49].

Nevertheless, the most frequently detected interdependence in our study indicates a greater-than-expected co-occurrence of predators and prey on the same day and/or at the same

crossing structure. Such temporal interactions have been found for small mammals with small mustelids, cats, and the Eurasian badger and for lagomorphs with both Eurasian badger and red fox. Similar overlaps have been described in marine ecosystems for spinner dolphins, *Stenella longirostris*, which appear to track the spatial and temporal patterns of their prey [50]. In general, studies that address the use of space by predators and prey from the perspective of the predators have focused on correlations between predator and prey densities but very rarely on the underlying behavioral mechanisms [36]. It is worth remembering that crossing structures may funnel animals of the surroundings towards them, thus increasing the chances of prey encountering predators even if they are at a low density [51].

The most notable of our conclusions is that predators and prey do not use wildlife passages independently and that observed dependencies seem related to size(prey)-matching, with potential repercussions for the usefulness of such structures, as was also recently pointed out by Caldwell and Klip [19]. On the one hand, crossing structures may become prey traps if predators use them to seek prey [16]. This possibility is consistent with studies that have considered the problem only in spatial terms and which reveal a correlated preference between certain predator species and their potential prey [8,12]. However, matching habitat selection considerations between predators and prey cannot be ruled out in these cases of spatial—but not temporal—coincidence, as suggested by evidence against the prey-trap hypothesis posed by Ford and Clevenger [18]. On the other hand, the reluctance of some species to use crossing structures because of predator presences may lead to such passages being avoided altogether [10]. The consequences in any case would be either wildlife passages becoming hazardous to prey species due to increased predation risk or rendered worthless for them due to predator avoidance [52–54]. A deeper understanding of the relevance of these processes will require more manipulative field studies, the massive use of digital photo- and video-recordings, and long-term population studies to complement observational approaches like ours based on species co-occurrence patterns.

From the point of view of conservation biology, the potential consequences of predator-prey interactions at wildlife passages are, at first sight, highly dependent on the conservation status of the species involved. Clearly, the options that crossing structures may become either prey traps or avoided sites makes them dangerous or useless for certain prey species, whether through increased mortality or through population fragmentation. The situation is especially delicate where prey species of conservation concerns are involved, as it happens in Australia with some marsupial species affected by introduced predators. Here, the absence of a co-evolutionary history between predators and prey leads the latter not to avoid predator odors [14,55,56]. Conversely, where predators are of a greater conservation concern, their attraction to wildlife crossings to seek prey may reduce their chances of being struck by vehicles [57], at least if they do not act also as roadkill scavengers [58,59]. With a wider scope, the key point would be how these interactions affect the whole ecosystem from a functional point of view. That is, how ecosystem processes around a permanent human-made perturbation differ from their historical conditions. These novel ecosystems, in terms of Hobbs and colleagues [60], can have new or altered species combinations as a result from population processes that may afterwards cascade through all ecosystem processes. For instance, genetic effects differ among species, with unknown long-term consequences, like the local extinction of some of them [61,62]. Additionally, changes in mammal populations like ground-dwelling rodents do have profound effects on ecosystem processes like soil perturbation, seed harvest, and plant succession [63]. Crossing structures are aimed to provide sufficient gene flow to prevent the genetic isolation of populations dissected by roads [64,65], but the analysis of their actual efficacy is still in its infancy, while the relevant effects can be subtle and only detectable in the long run.

Finally, it is possible that the predator-prey interdependencies detected at wildlife passages may just reflect the same interactions at larger landscape extents [16]. To evaluate their impacts, it would be necessary to know whether predation rates in surrounding areas are increased by the presence of such passages and whether or not the intensity of these effects is sufficient to alter the survival of

particular species. The only case study until present has shown no evidence of such changes among large mammals in the Trans-Canada Highway [18].

On applied grounds, could anything be done to minimize this possible consequence of wildlife passages? Assuming that the decision to install such crossings is the correct one to reduce the risks of population fragmentation [66,67], there are two approaches to address the problem. First, installing a relatively high density of crossings and making them wider should increase permeability in general [68,69], as well as dilute the risk that prey and predators may meet. Indeed, regulations in certain countries already require that crossing structures should occur every 500–1000 m along infrastructures and that they should be more than 20-m-wide [70]. Second, it seems appropriate to include structural complexity, such as logs, rocks, or piles of dead wood, within passages as a way of reducing the risk that a meeting between predator and prey will mean the death of the latter [71,72]. Refuges can have a stabilizing effect on predator-prey interactions [73]. In any event, our results make it clear that the "road ecosystem" is an arena for predation-prey interactions among mid- and small-sized vertebrates, even if we do not yet have enough information to fully evaluate its importance.

**Supplementary Materials:** The following materials are available online at http://www.mdpi.com/1424-2818/12/4/166/s1: Figure S1: Study area location, Table S1: Characteristics of the monitored crossing structures: type, location and dimensions., Table S2: Model comparison and probabilities associated to parameters φ and/or δ included in the most informative models for each predator-prey species-pair.

**Author Contributions:** Conceptualization, C.M. and J.E.M.; methodology, J.H. and C.M.; software, C.M.; formal analysis, C.M. and J.E.M.; investigation C.M. and J.E.M., writing—original draft preparation, C.M. and J.E.M.; writing—review and editing, C.M., J.E.M., and J.H.; project administration, J.E.M.; and funding acquisition, J.H. and J.E.M. All authors have read and agreed to the published version of the manuscript.

**Funding:** This study forms part of the CENIT-OASIS project funded by a consortium supported by the Centro para el Desarrollo Tecnológico e Industrial, CDTI, of the Spanish Ministry of Science and Innovation. Field data were gathered as part of a research arrangement between the Universidad Autónoma de Madrid, the Environment Ministry, and the Centro de Estudios y Experimentación, CEDEX. The Comunidad de Madrid supports the research group through the REMEDINAL TE-CM Research Network (P2018/EMT4338).

**Acknowledgments:** We are very grateful to I. Hervás for their assistance with data collection and to J. Hines for assistance with constructing the presence models. In memoriam to Francisco Suárez (1953–2010), who began and led this research area in our group.

**Conflicts of Interest:** The authors declare no conflicts of interest.

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
