# Peer review of "Attraction and Avoidance between Predators and Prey at Wildlife Crossings on Roads"

_diversity, doi:10.3390/d12040166_

Round 1

Reviewer 1 Report

  • While cooccurrence spatially or temporally is suggestive of potential predator-prey interactions, it does not provide definitive evidence these interactions occur. For this, video recordings and field manipulation of predators/prey and their cues would be more appropriate. In the discussion, it would be nice if authors address the need for more manipulative field studies and studies employing video recordings in addition to cooccurrence models.
  • Authors demonstrate crossing usage patterns deviate from chance (i.e. spatial interdependence), however, under natural conditions (i.e. areas where roads and associated crossing structures are absent) what is the normal likelihood of cooccurrence? How does this affect your models?
  • While authors demonstrate several instances of temporal cooccurrence among predator and prey types, monitoring occurred over 24 hours. Are authors comfortable concluding that predator/prey interactions (i.e. avoidance/attraction) occur when there is a potential > 12 hour difference in passage usage between predator and prey types? The methods section would benefit from more justification of sampling time frame.
  • In the methods section it would be good if authors explained why only 1 m strips of marble dust were used and no other additional methods (e.g. video recordings).
  • Authors have not sampled control locations (e.g. places not near crossing structures). For this reason, authors cannot conclude that the patterns of cooccurrence observed are different from areas where roads and wildlife crossings are absent. This is a major criticism that needs to be addressed throughout this manuscript.
  • Methods: Were the 1m wide strips of marble dust left for 24 hours and then checked early in the morning? It’d be great to specify the time frame (in hours) ‘dust traps’ were set for, in addition to the fact dust traps were checked early in the morning.
  • Methods: it would be good to discuss in more detail the crossing structures monitored. For instance, how large are drainage culverts compared to underpasses? How likely do you think this affects their use by the species monitored? Perhaps a figure would relay these differences more effectively?
  • Figure S2 is confusing – if only 110 kms of the A-52 motorway was monitored, why does the x-axis go to 220 km and why do several crossing structure types appear after 110 km?
  • With respect to parameters j and d when does a value of </> 1 become ecologically meaningful? For instance, “temporal cooccurrence was greater than expected for the cat-small mammal pair (1.4%)” but d is 1.01(0.15). Given this value is so close to 1 does it really indicate meaningful cooccurrence? Is there any literature suggesting a cut off value? At this stage, I feel this is misleading/ e.g. akin to a significant relationship with low r-square.
  • An equally interesting question would be - does species cooccurrence, temporally or spatially vary within predator and within prey types? Perhaps something to discuss in the discussion?

Reviewer 2 Report

This paper provides an important analysis of the interaction between predators and prey at focal points provided by fauna under/over-passes.  The methodology has application in other locations around the world, is clearly described and the results that are generated can be interpreted easily and logically.

I have a few minor comments that the authors might wish to consider.

L112 Suggest this line be amended to read "parameters that allow testing for the ..."

L119 The word 'potential' should be singular.

L137 I suggest that Figure 1 be deleted entirely as it adds nothing more to the Methods than is already clearly provided in the text from L117-136.

L146 Suggest this line be amended to read "types by mustelids ...".

L189 The sentence "In these both pairs ..." should be reworded to say "In both these pairs ...".

L249 The word 'These' should be replaced with "they" or "these species".

L279 "... evidences against..." should read "... evidence against ..."

L284 The authors have taken a narrow view that predator-prey interactions have most significance when either the predator or prey species are of conservation significance.  This ignores the ecological functions that individual species perform and how the loss of prey species due to predation, or their avoidance of some habitat due to the presence of prey species reduces the nature and extent of key ecological actions such as seed dispersal, biopedturbation and the associated effects on soil water permeability and nutrient distribution.  Over time this will lead to smaller isolated patches of habitat becoming unsuitable for some prey species entirely. 

The authors also make no comment on the effects that such changes in the movement of prey species between habitats transected by road/rail infrastructure have on the genetics of populations either side of the infrastructure.  The authors might like to read Ramalho et al. (2018)PLoS ONe 13(2): e0191190. and some of the other references included there-in to get an understanding of how important disruption of gene-flow might also be in these types of modified landscapes.

Finally, there seems to be a general lack of consistency in the formatting of the references, with some of the reference titles capitalized and others not.  I suggest the authors re-examine the Author Instructions for referencing. 
